

# Physiological response mechanism of *Machilus faberi* Hemsl under drought stress and rewatering

Wei Tang[1,*], Qiong Mo[1,*], Yangyang Fu[1], Damao Zhang[1], Yang Liu[1], Mingtong Ren[1], Tingting Li[1], En Wu[1], Dingding Su[2], Xiaoying Yu[1], Lihong Yan[3] and Yanlin Li[1,2,4]

[1] Engineering Research Center for Horticultural Crop Germplasm Creation and New Variety Breeding (Ministry of Education), Hunan Mid-Subtropical Quality Plant Breeding and Utilization Engineering Technology Research Center, Hunan Agricultural University, Changsha, China
[2] Institute of Advanced Agricultural Sciences, Peking University, Weifang, China
[3] Hunan Botanical Garden, Changsha, China
[4] Yuelushan Laboratory, Changsha, China
[*] These authors contributed equally to this work.

Corresponding authors
Lihong Yan, yanlh0424@163.com
Yanlin Li, liyanlin@hunau.edu.cn

## ABSTRACT

Drought stress is a predominant environmental challenge that significantly limits plant growth and survival, particularly affecting agricultural productivity and ecological stability in arid and semi-arid regions. This study aimed to elucidate the physiological responses of one-year-old *Machilus faberi* Hemsl seedlings to various degrees of drought stress, thereby aiding their cultivation and application in challenging environments. To simulate real-world drought conditions, four levels of drought stress were defined based on soil moisture content: control (80% soil moisture), mild drought (LD, 50–60% soil moisture), moderate drought (MD, 40–50% soil moisture), and severe drought (SD, 20–30% soil moisture). These thresholds were selected to represent a gradient from optimal water availability to extreme water scarcity, reflecting conditions commonly encountered in drought-prone regions. The results revealed that drought stress profoundly inhibited growth, primarily affecting plant height and reducing the number and length of new shoots. Notably, leaves under moderate and SD conditions demonstrated significant wilting and subsequent death. Photosynthetic pigment content and photosynthesis-related parameters initially increased but subsequently experienced a sharp decline as drought stress persisted. Biochemical analyses indicated elevated levels of relative conductivity and malondialdehyde, indicating extensive cell membrane damage. In the meanwhile, the activities of key antioxidant enzymes (superoxide dismutase, peroxidase, and ascorbate peroxidase) increased, alongside higher accumulations of soluble sugars, soluble protein, and proline, albeit with a sluggish recovery observed under severe stress conditions. Anatomical studies highlighted the thickening of both the upper and lower epidermis, as well as a reduction in the density of palisade and spongy tissues. Recovery following rewatering was more effective under LD and MD conditions than under SD stress, indicating that *M. faberi* possesses strong drought tolerance but is not suitable for highly arid regions. This study elucidates the adaptive mechanisms of *M. faberi* under drought stress and provides practical guidance for its management and cultivation in drought-prone areas, enhancing its ecological and economic viability.

## INTRODUCTION

Drought stress is one of the most significant abiotic challenges limiting plant growth and productivity, particularly in arid and semi-arid regions where water scarcity exacerbates agricultural instability and ecological degradation (*Yan & Suo, 2012*). Intensified by global climate change, the increasing frequency and severity of drought events have put nearly one-third of the world's land at risk (*Vicente-Serrano et al., 2022*; *Xu et al., 2023*). This pressing challenge underscores the urgency for research on plant adaptive mechanisms to drought stress.

Drought stress primarily manifests as phenotypic changes in plants, including significant reductions in shoot growth and plant height. Due to insufficient water supply, the plant's new shoots cannot elongate normally, resulting in growth stagnation or slowing down, and the overall height decreases significantly (*Wang et al., 2024*). This growth inhibition phenomenon is a strategy for plants to cope with drought stress by reducing growth to lower water demand (*Salehi-Lisar & Bakhshayeshan-Agdam, 2016*). In addition, drought stress can also cause leaf curling, discoloration and early shedding, further affecting the photosynthetic capacity and overall health of plants. Drought stress also significantly affected the photosynthetic pigment content and photosynthetic efficiency of plants. Water deficit impairs the water balance in plants. To reduce water loss, plants close their stomata, however, this also limits carbon dioxide absorption and reduces photosynthetic efficiency (*Zhang et al., 2011*). At the same time, these changes not only affect the growth and development of plants, but also may lead to a decrease in the biomass and yield of plants (*Seleiman et al., 2021*). Under drought stress, the accumulation of reactive oxygen species (ROS) in plants is significantly increased, resulting in oxidative damage to cell membrane lipids, proteins and DNA (*Qamer et al., 2021*). In order to cope with oxidative stress caused by drought, the antioxidant defense system of plants is significantly enhanced, including the activity of antioxidant enzymes such as superoxide dismutase (SOD), catalase (CAT) and peroxidase (POD), to reduce ROS accumulation and oxidative damage of cells (*Cruz de Carvalho, 2008*). This enhanced antioxidant capacity is an important mechanism for plants to protect their cell structure and function under drought conditions. The effects of drought stress on plants are multifaceted, involving phenotypic, physiological and biochemical aspects, and these changes reflect the complex regulatory mechanisms adopted by plants to adapt to arid environments. Although some progress has been made in improving plant drought tolerance through traditional breeding techniques (*Gonzalez, 2023*), the response mechanisms of plants under drought stress such as the changes of photosynthetic efficiency, the dynamics of antioxidant enzyme activity and the recovery mode after rewatering are still insufficiently studied. Therefore, exploring these aspects will help to develop more targeted strategies to improve the drought tolerance of *Machilus faberi* Hemsl (*M. faberi*).

*Machilus faberi* Hemsl (*M. faberi*), a broadleaf evergreen tree of the Lauraceae family, mainly distributed in Yunnan and Sichuan of China (*Zhu et al., 2022*). Renowned for its high-quality timber, which is widely used in the furniture industry and also possesses medicinal value. Additionally, its graceful tree shape and dense canopy make it an excellent choice for garden landscaping, particularly as a courtyard ornamental tree. Although it has important economic and application value, the existing research has primarily focused on cultivation techniques (*Chen, 2013*), genetic diversity (*Zhu et al., 2022*), extraction of oil and fragrance compounds (*Yang et al., 2024*). So far, the specific physiological and biochemical responses of *M. faberi* to different degrees of drought stress remain elusive. The objective of this study is to systematically investigate the response of one-year-old *M. faberi* seedlings to drought stress and the physiological and biochemical responses after rewatering by detecting a series of comprehensive physiological, biochemical and anatomical parameters, so as to provide scientific basis for their drought-resistant strategies. These results will reveal the dynamic pattern and the correlation of these parameters of *M. faberi* in response to different drought conditions and rewatering conditions, and extend the understanding of physiological and biochemical mechanism in these processes standing these mechanisms. Taken together, these observations will improve the adaptability of this species to climate change and its application in urban greening and management, especially in arid regions.

## MATERIALS & METHODS

### Plant materials and treatment

In May 2022, one-year-old seedlings of *M. faberi* with basically the same growth vigor were selected from the Germplasm Resources Garden of Hunan Botanical Garden Conservation Institute. The *M. faberi* seedlings was planted in a flowerpot (25 cm × 30 cm), using a cultivation substrate composed of garden soil, peat, perlite, and cow dung at a ratio of 8:6:3:1. The experimental site was located in the greenhouse of the flower base. In the greenhouse of the flower seedling base, all *M. faberi* seedlings were uniformly managed. The temperature was maintained at 25 °C during the day and 20 °C at night, with a relative humidity of 65% (temperature and humidity were measured using temperature and humidity sensors (DHT22, Shenzhen, China)). The light conditions were set to a 14-hour light/10-hour dark cycle with a light intensity of 500 $\mu$mol m$^{-2}$ s$^{-1}$.

After one month of pre-culture, different drought treatments were carried out using pot water control method (*Wang et al., 2024*). Water control treatments were implemented using a gravimetric method, wherein soil moisture content was monitored and maintained by regularly weighing of soil samples. The soil moisture content (%) was calculated as: Soil moisture content $= \frac{\text{Wet Weight}-\text{Dry Weight}}{\text{Dry Weight}} \times 100\%$. This calculation reflects the proportion of water in the soil relative to its dry weight. Four distinct soil moisture regimes were established to simulate different drought intensities: control (CK, soil moisture content 80%, representing well-watered conditions), mild drought (LD, soil moisture content 50%–60%), moderate drought (MD, soil moisture content 40%–50%), and severe drought (SD, soil moisture content 20%–30%). Soil moisture levels were adjusted and maintained daily by adding the required volume of water to compensate for losses due to

evapotranspiration, ensuring consistent treatment conditions throughout the experimental period. Each replicate consisted of five pots, and the entire experiment was repeated three times for consistency. After the drought treatment, the plants were watered daily for seven days and this process was repeated three times to simulate the recovery phase. Rewatering was applied with a controlled amount of water to ensure consistent soil moisture. Data were collected on days 7, 14, and 21 after rewatering (WT7, WT14, and WT21). For each sampling date, leaf samples were taken from different plants to avoid potential confounding effects of leaf removal on the measured parameters.

The entire experiment comprised two phases: drought stress treatment (T0, T7, T14, T28, T35, T42, T49 and T56) and rewatering (WT7, WT14 and WT21). At 10:00 a.m. on the corresponding day of each treatment group, five mature leaves were collected from each plant and immediately transported to the laboratory for subsequent analysis, including the determination of chlorophyll content, malondialdehyde (MDA) concentration and antioxidant enzyme activity.

## Morphological data measurement

The morphological characteristics measured included plant height, shoot growth length and shoot number. Plant height was measured using a tape measure from the soil surface of the potted plant to the top of the main stem. The shoots refer to the new growth emerging from the main stem after pruning. The length of all new shoots on each branch was measured and the average length was used as a replicate.

## Chlorophyll content measurement

Mature leaves were collected at 10:00 a.m. on sunny days from the top of the plant and washed with deionized water. The samples were dried and sectioned on transparent paper. Then, 0.2 g of fresh leaf samples were weighed and soaked in 10 ml 95% ethanol for 24 h under dark conditions. Absorbance at 470 nm, 649 nm, and 665 nm was determined using an ultraviolet spectrophotometer (UNICO, Shanghai, China) (*Dai et al., 2023*). This process was repeated during drought stress treatment (T0, T7, T14, T21, T35, T42, T49 and T56) and rewatering (WT7, WT14 and WT21) to monitor changes in chlorophyll content.

## Determination of photosynthetic parameters

Photosynthetic parameters were measured using a portable photosynthesis system (LI-6400, LI-COR Biosciences, Lincoln, NE, USA) under controlled conditions. Measurements were conducted between 9:00 and 11:30 a.m. on sunny days to minimize diurnal fluctuations in photosynthetic activity. Five mature leaves were selected randomly from each treatment group, ensuring uniformity in developmental stage and exposure. The maximum net photosynthetic rate (Pn), stomatal conductance (Gs), transpiration rate (Tr) and intercellular $CO_2$ concentration (Ci) were recorded at a controlled light intensity of $500\ \mu mol\ m^{-2}\ s^{-1}$, which is commonly used in such studies to represent moderate light conditions. This light intensity avoids potential light saturation or photoinhibition effects and ensure an accurate assessment of photosynthetic response. The assays were taken from a six $cm^2$ area of a leaf from a one-year-old *M. faberi* seedling, and the leaves were allowed to stabilize under the set conditions for approximately 2 min before data recording. Each

measurement was repeated three times to ensure consistency and reliability, following standard protocols.

## Determination of relative conductivity and MDA

The plant leaves of the same size were selected and washed three times with distilled water. The surface water was removed using filter paper. The leaves were then cut into strips of suitable length (avoiding the main vein), and 0.1 g fresh samples were weighed in triplicate and placed in a centrifuge tube containing 10 ml of deionized water. The samples were soaked for 12 h at room temperature, and then the electrical conductivity of the extract (R1) was measured using a conductivity meter (DDS-11A, Shanghai, China). After that, the samples were heated in a boiling water bath for 30 min, cooled to room temperature and shaken, and then the electrical conductivity of the extract (R2) was measured again (*Chen et al., 2010*). Relative conductivity was calculated as the ratio of R1 to R2 (R1/R2).

For MDA assay, 0.1 g sample was added to a solution containing 0.1% trichloroacetic acid, centrifuged at 10,000 g for 10 min, and 500 μl of supernatant was collected. Then 500 μL of 20% trichloroacetic acid solution containing 0.5% thiobarbituric acid (TBA) was added and heated at 90 °C for 20 min. After ice bath, the solution was centrifuged at 10,000 rpm/min for 5 min. The absorbance of the supernatant was measured at 532 nm and 600 nm, and the 20% trichloroacetic acid solution containing 0.5% TBA was used as a control (*Zhang et al., 2016*).

## Detection of antioxidant system

The activity of SOD was determined by NBT reduction method, POD activity was determined by guaiacol colorimetric method, APX activity was determined by UV spectrophotometry (*Luo, 2021*), and three groups of repeated controls were set up for each determination.

## Detection of osmotic adjustment system

The content of soluble sugar was determined by anthrone-sulfuric acid colorimetry. The content of SP was determined by coomassie brilliant blue method. The content of Pro was determined by acid ninhydrin method (*Wang et al., 2023*). Three replicates were set for each determination.

## Observation of leaf anatomical structure

An inverted microscope was used to observe and photograph the leaves of *M. faberi* that had been made into paraffin sections, and Leica Application Suite V4 was used to measure the numerical differences of anatomical structures (upper epidermis, lower epidermis, palisade tissue, sponge tissue) of *M. faberi* leaves under different treatments.

## Data analysis

All experiments were performed with three biological replicates, and the results are presented as the mean ± standard deviation of three independent experiments. The plant height, new branch number, new branch length, chlorophyll a, chlorophyll b, chlorophyll a+b, chlorophyll a/b, Pn, Ci, Tr, RC, MDA content, SOD activity, POD activity, APX activity, SP content, SS content and Pro content were analyzed using one way analysis of
variance (ANOVA) with SPSS software. Duncan's multiple range test was employed to determine significant differences between means when the ANOVA results were significant. A significance level of $p < 0.05$ was used to define statistically significant differences.

FactoMineR and ggplot2 packages of R was used to do principal component analysis and correlation analysis of plant height, new branch number, new branch length, chlorophyll a, chlorophyll b, chlorophyll a+b, chlorophyll a/b, Pn, Ci,Tr, RC, MDA content, SOD activity, POD activity, APX activity, SP content, SS content and Pro content. All the data were obtained from different time points under drought stress, and each treatment group carried out independent repeated experiments (three biological replicates in each group). To ensure the reliability of the correlation analysis, all data were normalized relative to the control plants at each corresponding time point before statistical analysis. Specifically, the normalization process involved subtracting the mean value of the CK group at the corresponding time point from the treatment group data, followed by scaling based on the standard deviation of the CK group. The Pearson correlation coefficients were calculated using the mean values of three independent biological replicates per group. Additionally, variance analysis was performed within the CK group to confirm that fluctuations were within acceptable experimental limits and did not introduce artefacts into the correlation results.

## RESULTS

### Effects of leaf phenotype and growth under drought stress

The effects of different levels of drought stress on leaves showed evident differences (Fig. 1A). Under LD stress, leaves started to drop on the 56th day, characterized by a noticeable loss of turgor and partial curling. Under MD stress, leaves exhibited visible wilting by the 42nd day, accompanied by yellowing and the necrosis of older leaves. SD stress led to pronounced shrinkage and withering of leaves by the 36th day, with a rapid progression to the desiccation and abscission of older leaves. After rewatering, plants subjected to LD and MD stress demonstrated significant recovery, regaining turgor and chlorophyll integrity, whereas recovery in the SD group was slower and incomplete (Fig. 1B).

Moreover, different degrees of drought stress also significantly affected the overall growth of plants. During the whole drought treatment, the new shoot length and plant height of each treatment group increased slowly. After rewatering, these parameters increased rapidly, except for the plant height of the SD group, which decreased in WT14-WT21 (Figs. 1C–1D). Under drought stress, compared with CK group, the number of new shoots of *M. faberi* leaves in each treatment group decreased from T0 to T14, and then fluctuated (Fig. 1E).

### Effects of photosynthetic pigments under drought stress and rewatering

On the 21st day of drought stress, the contents of chlorophyll a and chlorophyll b in *M. faberi* leaves increased significantly, reaching a peak. Specifically, under different drought treatments, the content of chlorophyll a ranges from 2.41 to 2.60 mg/g (Fig. S1A), while the

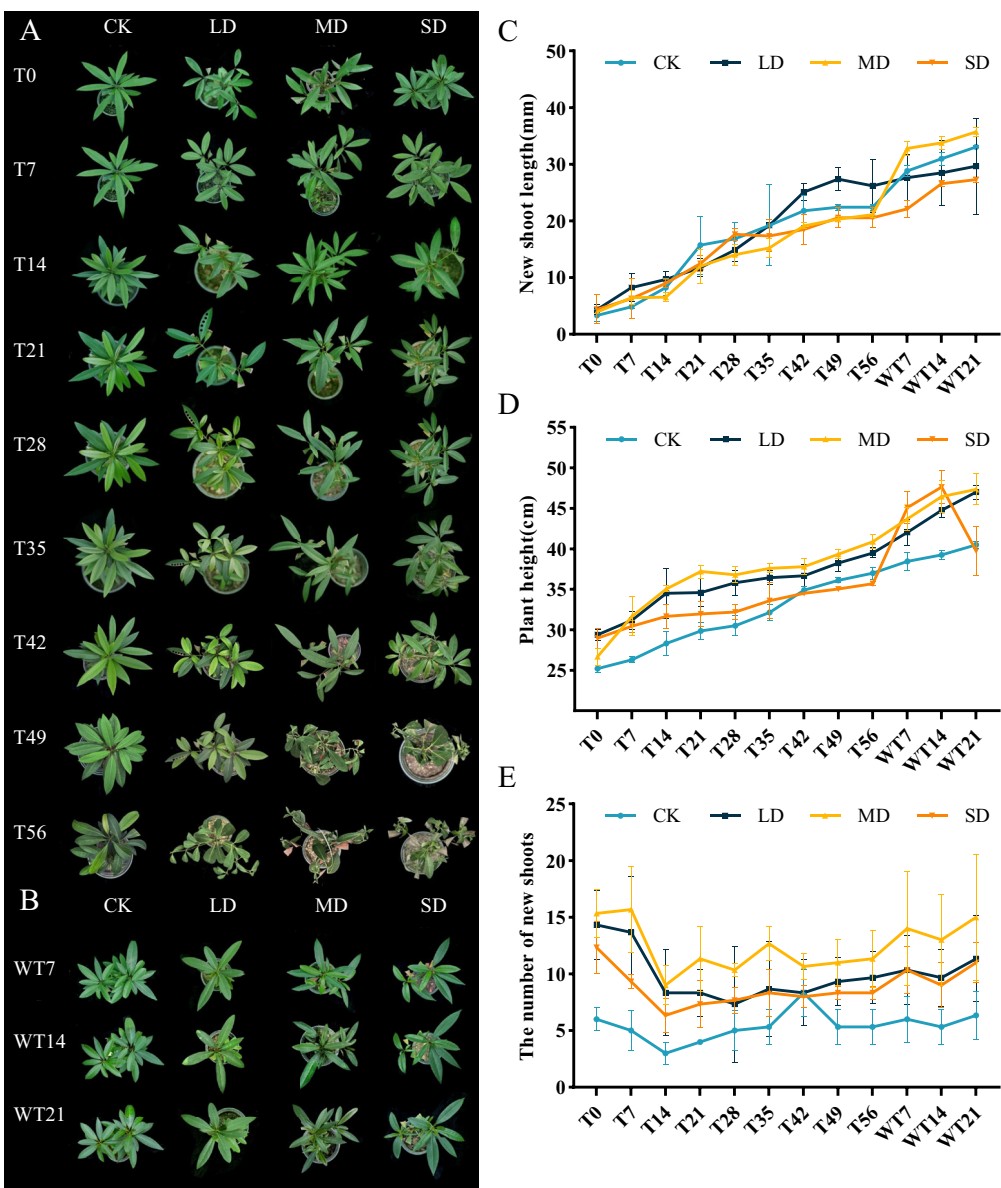

**Figure 1** **Different degrees of drought stress on phenotype and morphological index of *M. faberi* at 0, 7, 14, 21, 28, 35, 42, 49, 56 days and rewatering (WT7, WT14, WT21).** (A) Phenotypic changes under drought treatment; (B) Phenotypic changes under rewater treatment; (C) New shoot length; (D) Plant height; (E) The number of new shoots. Note: CK, soil moisture content 80%; LD, soil moisture content 50%–60%; MD, soil moisture content 40%–50% and SD, soil moisture content 20%–30%.

content of chlorophyll b ranges from 1.01 to 1.15 mg/g. However, with the intensification of drought stress, chlorophyll content began to decline. After rewatering, chlorophyll a and chlorophyll b levels recovered significantly regardless of the drought stress level, and exceeded the CK group after three times of rewatering.

Under SD stress, the chlorophyll a/b ratio continued to decline until the 28th day. Under MD stress, the chlorophyll a/b ratio was relatively stable and reached the minimum on the

35th day. In contrast, under LD stress, the chlorophyll a/b ratio continued to rise until the 49th day, reaching a maximum of 2.47 mg/g (Fig. S1C). The chlorophyll a+b showed an upward trend on the 21st day of drought stress, followed by a decline until the 56th day. After rewatering treatment, the chlorophyll a+b increased again and exceeded the levels in the CK group (Fig. S1).

## Effects of photosynthetic parameters under drought stress and rewatering

After drought treatment, the Ci of the leaves decreased and then increased. LD and MD groups reached the minimum on the 35th day, and SD group reached the minimum at T21 period. Then the Ci of each treatment group increased, but was lower than that of the control group. After rewatering, the Ci of LD and MD treatment groups began to increase significantly, while the SD treatment group was still in the mean stress period (Fig. 2A).

Under the whole drought treatment, the Pn of LD and MD groups showed a trend of increasing first and then decreasing under continuous stress, while the SD group showed a gradual downward trend (Fig. 2B). After rewatering, the Pn gradually increased. The recovery rates under different degrees of stress were as follows: LD > MD > SD, with all recovery levels significantly lower than that of the CK group.

Under different stress levels, the Gs of *M. faberi* leaves showed different patterns. The LD and MD groups showed a trend of increasing first and then decreasing under continuous stress, while the SD group showed a gradual downward trend. After rewatering, the Gs of LD and MD treatment groups increased, while that of SD group was lower than other group in WT7 period (Fig. 2C).

During drought stress, all groups showed lower Tr than To. During drought stress, there was no significant difference in Tr between CK and all stressed group until T14 (Fig. 2D). The differences compared to CK appeared at T21, T28 T35, T42, T49, T56, WT7 and WT14 for MD and SD groups, and exhibited at T35, T42 and T56 for LD group (Fig. 2D). After rewatering, all groups recovered well at WT21 (Fig. 2D).

## Effects of relative conductivity and MDA content under drought stress and rewatering

The RC of *M. faberi* leaves showed significant changes across different treatment groups (Fig. 3A). Under LD treatment, the RC initially increased, then decreased on the 28th day, increased again to a peak on the 49th day, and finally decreased on the 56th day. Compared with CK group, LD group showed significant difference at T7, T42 and T56, MD group exhibited significant difference at T35, T42, T49 and WT7, while SD group displayed significant difference at T21, T42, T49, T56, WT7 and WT14 (Fig. 3A). After rewatering, all groups showed no significant difference at WT21 (Fig. 3A).

The MDA content in all treatment groups initially increased and then decreased as drought stress duration extended, but the MDA content differed significantly across treatments. Under LD stress, it peaked at 36.37 nmol/g FW on the 21st day, significantly higher than the CK group at T7, T21, T56 and WT7, but significantly lower than the CK group at WT14 and WT21 (Fig. 3B). Compared with CK group, MD group showed

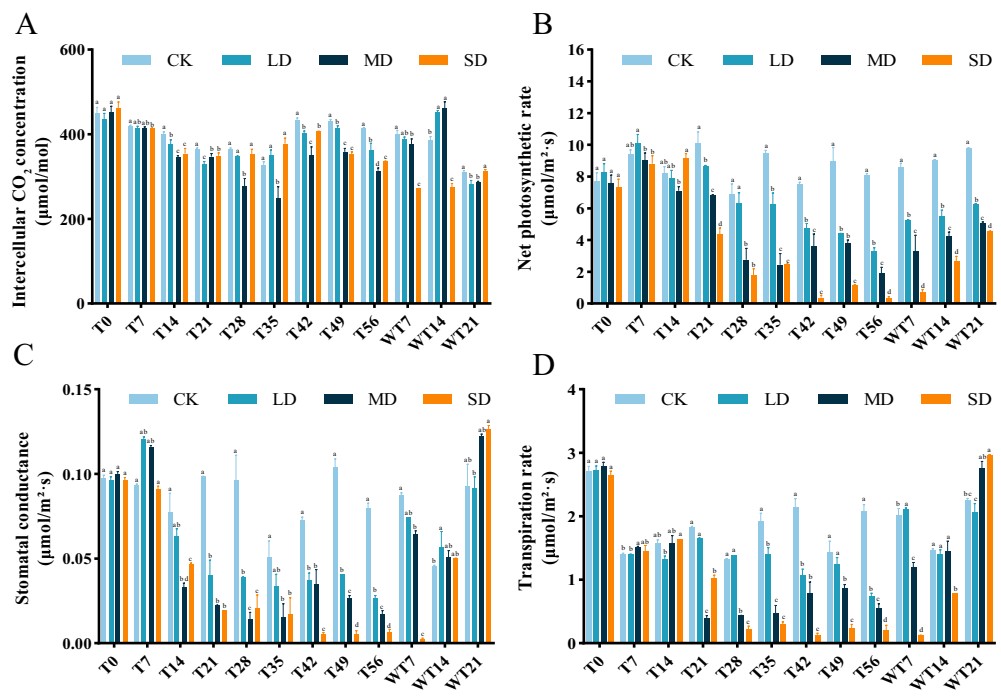

**Figure 2** Different degrees of drought stress on photosynthetic parameters of *M. faberi* at 0, 7, 14, 21, 28, 35, 42, 49, 56 days and rewatering (WT7, WT14, WT21). (A) Intercellular CO2 concentration; (B) Net photosynthetic rate; (C) Stomatal conductance; (D) Transpiration rate. Bars with different lowercase letters (a, b, c, d) indicate significant differences between groups at $P < 0.05$. Note: CK, soil moisture content 80%; LD, soil moisture content 50%–60%; MD, soil moisture content 40%–50% and SD, soil moisture content 20%–30%.

significant difference at T7, T14, T21, T28, T35, T56, WT7 and WT14, while SD group exhibited significant difference at T7, T14, T21, T35, T42, T56, WT14 and WT21 (Fig. 3B).

## Effects of drought stress and rewatering on antioxidant system

Under continuous drought stress, POD activity all stressed groups were increased than To. Compared with CK group, LD group showed significant higher POD activity at T21, T28, T35, T42, T49 and T56, MD group exhibited significant higher POD activity at T7, T21, T28, T35, T42, T49 and T56, and SD group displayed significant higher POD activity at T7, T21, T28, T35, T42, T49 and T56 (Fig. 4A). LD and MD groups exhibited a gradual increase, reaching maximum values of 37,495.80 ($\times 10^3$ U/min/g FW) and 34,167.07 ($\times 10^3$ U/min/g FW) on the 56th day of stress, respectively, and they already started to differ from CK on T21. After rewatering, LD and MD groups showed higher POD activity than CK group at WT7, while LD, MD and SD groups exhibited significant lower POD activity than CK group at WT 14 and WT21 (Fig. 4A).

SOD activity also increased under drought stress, reaching a peak level of 202.10, 205.17, and 213.68 U/g FW on the 56th day in the LD, MD, and SD groups. Compared with CK group, LD group displayed significant difference at T7, T42, T49 and T56, MD group showed significant difference at T7, T42, T49, T56, WT14 and WT21, while SD group

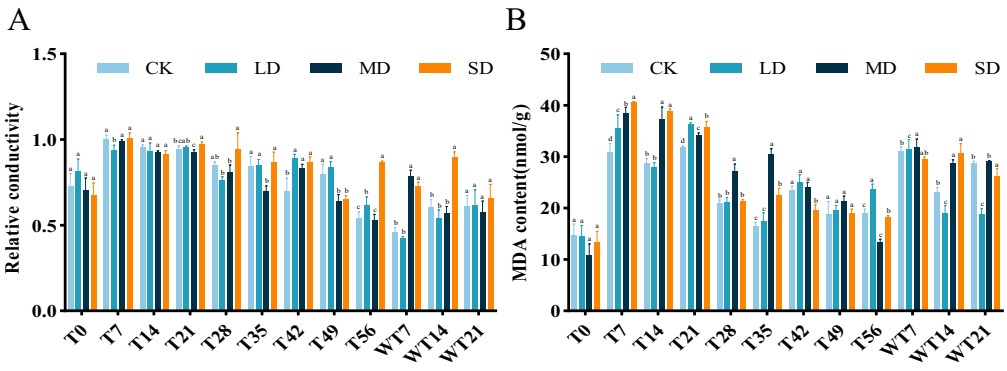

**Figure 3** **Different degrees of drought stress on relative conductivity and malondialdehyde of *M. faberi* at 0, 7, 14, 21, 28, 35, 42, 49, 56 days and rewatering (WT7, WT14, WT21).** (A) Relative conductivity; (B) Malondialdehyde content. Bars with different lowercase letters (a, b, c, d) indicate significant differences between groups at $P < 0.05$. Note: CK, soil moisture content 80%; LD, soil moisture content 50%–60%; MD, soil moisture content 40%–50% and SD, soil moisture content 20%–30%.

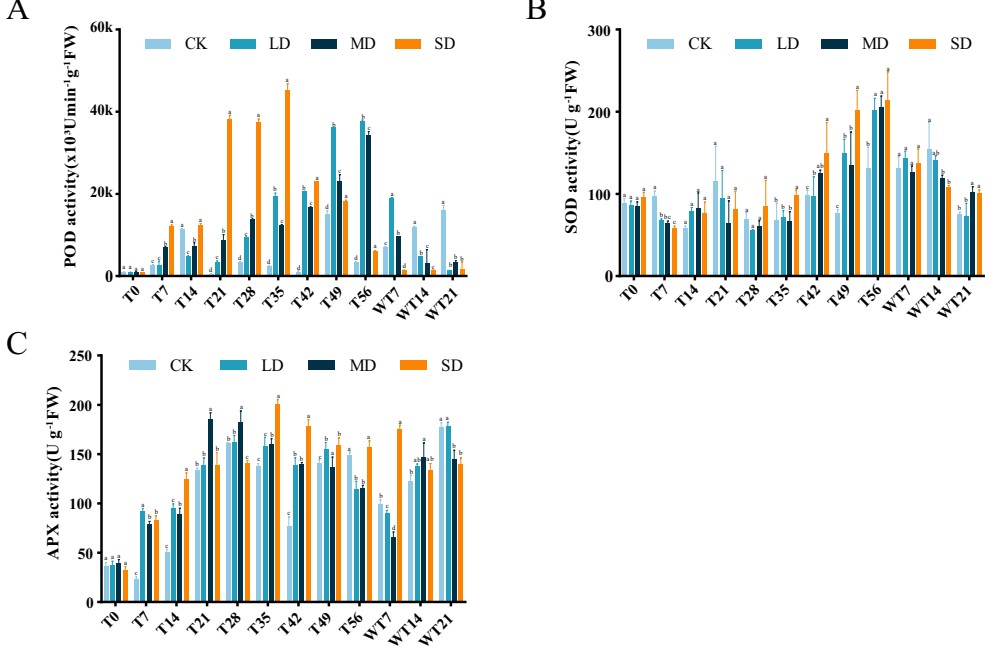

**Figure 4** **Different degrees of drought stress on antioxidant enzymes of *M. faberi* at 0, 7, 14, 21, 28, 35, 42, 49, 56 days and rewatering (WT7, WT14, WT21).** (A) Superoxide dismutase; (B) Peroxidase; (C) Ascorbate peroxidase. Bars with different lowercase letters (a, b, c, d) indicate significant differences between groups at $P < 0.05$. Note: CK, soil moisture content 80%; LD, soil moisture content 50%–60%; MD, soil moisture content 40%–50% and SD, soil moisture content 20%–30%.

exhibited significant difference at T7, T35, T42, T49, T56, WT14 and WT21 (Fig. 4B). After rewatering, SOD activity in all groups gradually decreased.

APX activity demonstrated an initial increase followed by a decrease under drought stress (Fig. 4C). The LD group peaked at 162.29 U/g FW on the 28th day, the MD group reached its maximum on the 21st day, and the SD group peaked at 200.48 U/g FW on the 35th day, respectively (Fig. 4C). In comparison to CK group, LD group displayed significant difference at T7, T14, T35, T42, T49 and T56, MD group showed significant difference at T7, T14, T21, T28, T35, T42, T49 and T56, and SD group exhibited significant difference at T7, T14, T28, T35, T42, T49 and T56 (Fig. 4C). After rewatering, APX activity decreased at WT7 and then increased at WT14 and WT 21 in CK, LD and MD groups, while APX activity decreased at WT14 and WT21 in the SD group (Fig. 4C). All stress treatment groups (LD, MD and SD) showed significant difference compared with CK group at WT7, MD displayed significantly higher APX activity than CK, while MD and SD exhibited significantly lower APX activity than CK and LD at WT21 (Fig. 4C).

## Effects of drought stress and rewatering on osmotic adjustment substances

The content of SS exhibited a general downward-upward-downward trend. During the stress period, the SS content of LD and MD treatment groups reached the maximum on the 28th day, which were 57.99 mg/g and 58.30 mg/g, respectively. The SD treatment group reached a maximum of 52.65 mg/g on the 7th day, significantly higher than the CK group (Fig. 5A). After rewatering, the SS content of each treatment group showed an increasing trend, reaching the highest value on the 7th day of rewatering, with significantly higher level at both WT7 and WT14 than CK group. At WT21, the SS content of LD and MD group was significantly lower than CK group, while SD group showed no significant difference with the CK group (Fig. 5A).

The results showed that the SP content of *M. faberi* increased gradually. Under LD stress, the SP content reached the highest value of 7.17 mg/g on the 28th day, significantly higher than that of the CK group at T28. Under MD and SD stress, the SP content reached the maximum of 9.48 mg/g and 9.41 mg/g on the 49th day, respectively, significantly higher than that of the CK group at T49 (Fig. 5B). After the end of drought stress, the plants returned to normal water supply, and the SP content of each treatment group began to decrease slowly.

Pro content exhibited an upward–downward–upward trend under LD treatment, with a peak of 5.65 mg/g on the 21st day and a minimum of 0.34 mg/g on the 49th day. Under MD stress, Pro content fluctuated, ultimately reaching a maximum of 7.96 mg/g on the 56th day, significantly higher than the CK group. Notably, under SD stress, Pro content increased sharply at T49, T56 and WT7, significantly higher than the CK, LD and MD groups. After rewatering, Pro content decreased in all treatments, with higher levels in LD group at WT21, in MD group at WT7, and in SD group at WT7 and WT14 (Fig. 5C).

## Effects of drought stress and rewatering on leaf anatomical structure

Anatomical structure of the leaves revealed that under LD stress, the thickness of the upper epidermis increased slowly, showing no significant overall upward trend compared to the

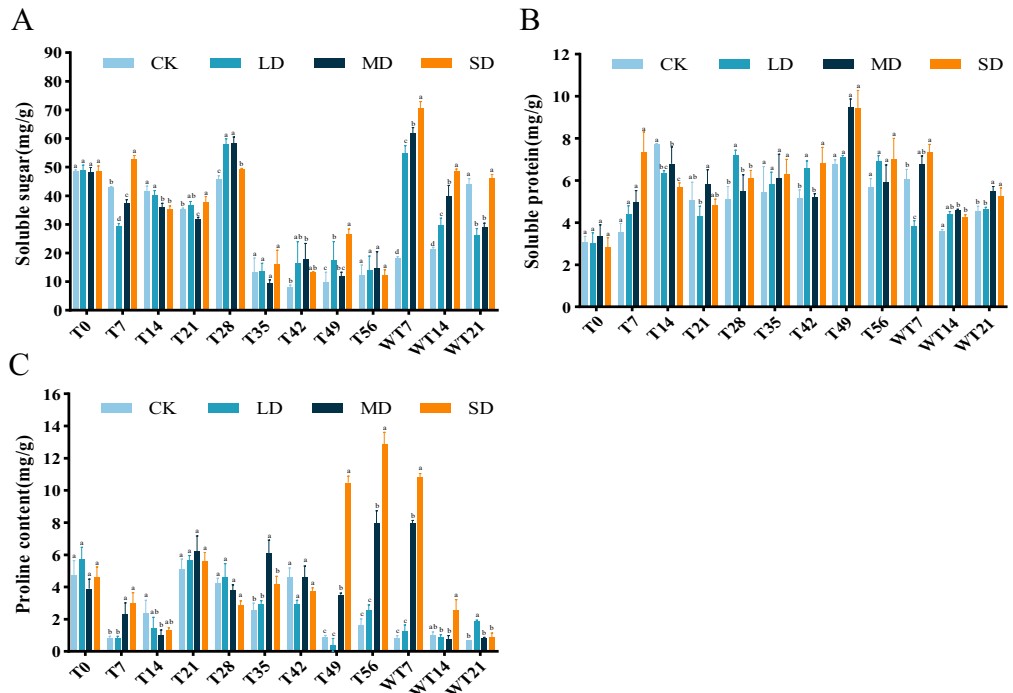

**Figure 5** *Different degrees of drought stress on osmotic adjustment substances of M. faberi at 0, 7, 14, 21, 28, 35, 42, 49, 56 days and rewatering (WT7, WT14, WT21).* (A) Soluble sugar; (B) Soluble protein; (C) Proline. Bars with different lowercase letters (a, b, c, d) indicate significant differences between groups at $P < 0.05$. Note: CK, soil moisture content 80%; LD, soil moisture content 50%–60%; MD, soil moisture content 40%–50% and SD, soil moisture content 20%–30%.

CK group. Under MD and SD stress, the thickness of the upper epidermis showed an upward trend, but the MD stress showed a downward trend from the 49th day until the 14th day of rewatering, and the SD showed that the thickness of the epidermis continued to decrease after the stress (Fig. S2A). Under LD stress, the thickness of palisade tissue continued to rise to the 28th day, and reached the highest value of 0.093 μm at this time. Then, with the extension of stress time, it began to gradually decrease until the normal watering after the stress, and the thickness of palisade tissue began to rise. The changing trend of palisade tissue under MD stress was consistent with that of LD, and the difference was that MD began to decrease until the 35th day. Under SD stress, the palisade tissue of leaves reached the lowest value of 0.05 μm on the 35th day, and then began to rise (Fig. S2B). The variation trend of sponge tissue under different drought degrees was inconsistent (Fig. S2C). The observation of the anatomical structure of the lower epidermis showed that the thickness of the lower epidermis of the leaves under LD showed an overall upward trend. Under MD stress, the thickness of the lower epidermis showed a downward trend and then an upward trend. The thickness of the lower epidermis of SD showed an upward trend and continued to rise after rewatering (Fig. S2D).

Investigation of leaf tissue under drought stress showed that under LD and MD conditions, cell structures remained clear, with small intercellular spaces and tightly

arranged cells, with minimal changes in cell water loss. In contrast, under SD stress, palisade tissue showed narrowing and significant water loss by the 35th day, continuing until the 56th day (Fig. S3). After rewatering, cells in the rehydrated tissues of LD- and MD-treated plants exhibited compact arrangement with minimal intercellular spaces and well-defined structural integrity, whereas SD-treated cells showed a slower recovery, characterized by large intercellular spaces and a more disorganized, loose cellular arrangement (Fig. S4).

## Correlation analysis of each index of *M. faberi* under drought stress

There was a significant correlation between various physiological and biochemical indices of *M. faberi* under drought stress (Fig. 6). Gs was significantly positively correlated with Pn and Tr, and the correlation coefficient was above 0.85. There was also a significant positive correlation between Pn and Gs, and the correlation coefficient was above 0.90. There was a significant positive correlation between chlorophyll a/b and chlorophyll a, chlorophyll a+b, and the correlation coefficient was above 0.90. Chlorophyll a+b was significantly positively correlated with Pn, Tr and new shoot length. Chlorophyll a content was significantly positively correlated with Pn and Tr. Chlorophyll a/b content was positively correlated with Tr.

At the same time, we also found that chlorophyll a+b and chlorophyll a was significantly negatively correlated with Pro, SP and Ci content. There was a significant negative correlation between SS and SOD activity. There was a significant negative correlation between Pn and Pro.

## Principal component analysis

Principal component analysis revealed that the first principal component had a variance contribution rate of 52.7%, primarily contributed by plant height, Ci, and RC. The second principal component, with a variance contribution rate of 16.9%, was mainly contributed by POD activity, Tr, and photosynthetic rate. The cumulative contribution rate of these two principal components reached 69.4% (Fig. S5). The first principal component highlighted the significant role of chlorophyll b, chlorophyll a, chlorophyll a+b, Ci, plant height, and RC in reflecting the response of *M. faberi* leaves to drought stress. Overall, as drought stress increased from group A to group D, the relative aggregation of each treatment group gradually became more discrete, indicating a positive correlation between the degree of stress and the growth indices of *M. faberi*.

## DISCUSSION

The growth and development of plants depend on adequate soil moisture supply, with drought stress being one of the most significant adverse conditions that can lead to plant mortality (*Munns, 2002*). In this study, after suffering from different levels of drought stress, *M. faberi* showed the phenomena of withered leaves, yellowing and falling off, but gradually recovered after rewatering. This phenomenon may be a strategy where plants limit transpiration by reducing exposed surface area through leaf curling under mild water deficit conditions. Additionally, leaf curling provides partial shading, reducing ROS production in shaded areas and achieving water retention (*Picotte et al., 2007*; *Rueda, Godoy*

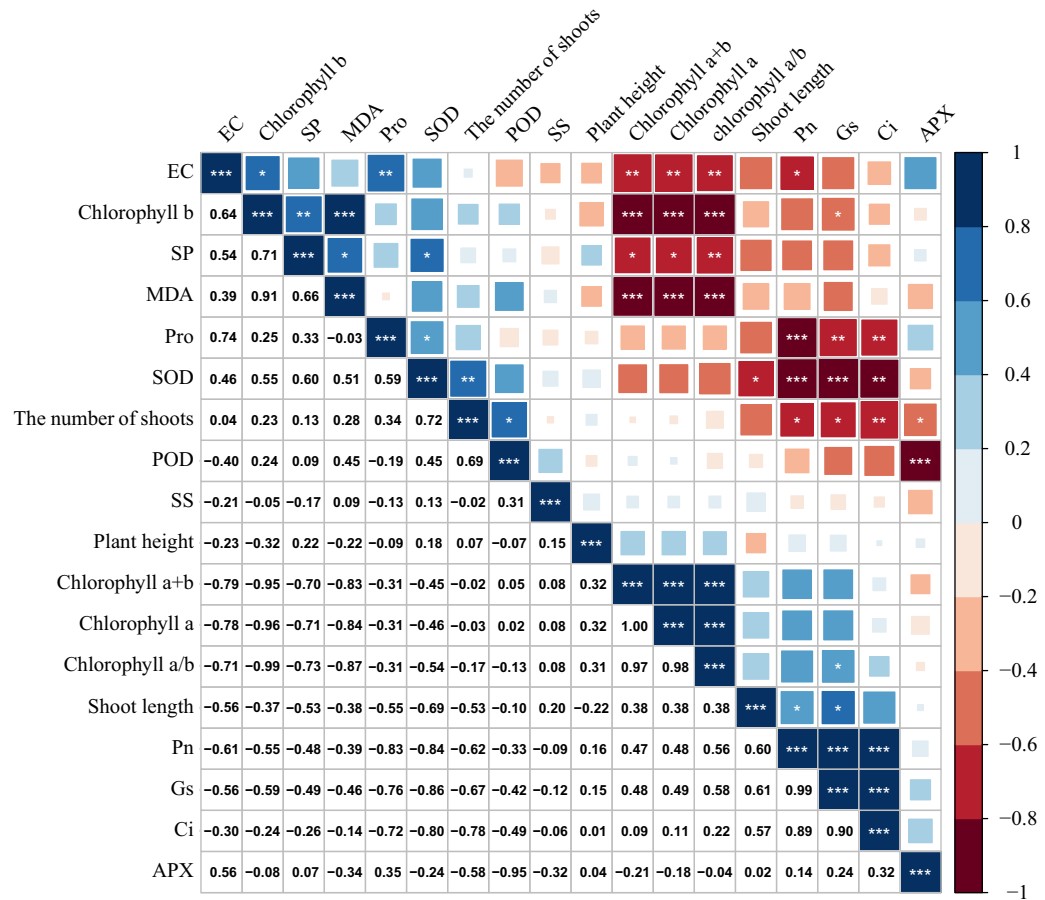

**Figure 6** Correlation analysis of *M. faberi* under drought stress and rewatering treatments. The number represents Pearson correlation coefficient, and asterisk represents the significance of Pearson correlation coefficient. (*) indicates $p < 0.05$, (**) indicates $p < 0.01$, and (***) indicates $p < 0.001$.

*& Hawkins, 2016*). Meanwhile, with the extension of drought time, plant height and new branch length of *M. faberi* decreased gradually. Compared with CK group, MD and SD stress significantly inhibited its growth, which is consistent with the findings in *Dalbergia odorifera* (*Jia et al., 2013a*) and *Excentrodendron hsienmu* (*Ou et al., 2017*). Drought leads to a decrease in soil moisture and difficulty in water absorption by plant roots, resulting in the slow growth of aboveground parts (*Mahmood et al., 2025*). The significant decrease in the number of new shoots at T14 further illustrates this view. When drought stress was alleviated through rewatering, the increase of plant height and shoot length was also recovered, indicating that plant growth was restored after rewatering, consistent with the previous studies (*Shi et al., 2023*; *Zhang et al., 2014*).

Most studies have shown that drought stress significantly inhibits photosynthetic capacity, primarily due to factors such as reduced stomatal conductance and diminished photosynthetic pigment synthesis (*McCree, 1986*). In this experiment, the content of chlorophyll a and b showed an initial increase and then a downward trend under drought

stress. This fluctuation may reflect a mechanism where plants maintain photosynthesis through increased pigment synthesis during early stress stages (*Chaves, Flexas & Pinheiro, 2009*). The chlorophyll content of each treatment group showed an upward trend in the early stage of drought (T0-T21), and then gradually decreased. This finding is consistent with the results observed in Chinese onions (*Guo et al., 2024*), which suggests that plants initially responded to drought stress by increasing chlorophyll content to enhance photosynthetic efficiency. After rewatering, the chlorophyll content gradually recovered, indicating that plants accelerated pigment synthesis after rewatering to compensate for pigment loss during drought, which is consistent with the results of Sang et al. (*Sang, Ma & Chen, 2011*). Meantime, in the study of *Populus alba* × *Populus berolinensis*, *Wang et al. (2006)* also proved this. However, with the aggravation of drought stress, the photosynthetic system was destroyed, resulting in a decrease in chlorophyll synthesis. This finding is consistent with the results observed in Chinese onions (*Guo et al., 2024*). However, the chlorophyll content of MD and SD groups still decreased sharply at the initial stage of rewatering (WT7), indicating that with the aggravation of drought stress, the photosynthetic system was destroyed and it was difficult to recover through short-term rewatering. This requires that in the conservation and management of *M. faberi* seedlings, it should be avoided to suffer from moderate or above drought stress.

*Fu et al. (2006)* showed that under mild and moderate drought stress, stomatal closure led to decreased net photosynthetic rate in *Populus pseudo-simonii* leaves, while severe drought caused structural and functional damage to photosynthetic organs. In this study, Pn, Tr, Ci and Gs of each treatment group were significantly reduced during T0-T35, indicating that stomatal closure reduced Tr and Pn at this stage. Despite stomatal closure, photosynthesis was still ongoing, which leads to partial consumption of $CO_2$ in leaves, which in turn reduces its content, which is consistent with the drought stress response mechanism of different plants (*Flexas et al., 2004*). We also found that although Pn, Gs and Tr decreased continuously, the intercellular $CO_2$ concentration in T35-T56 was higher than that in T35, indicating that at this stage, the factor affecting the change of photosynthetic parameters was not stomatal closure, but chloroplast damage, which was verified in many research results (*Lin et al., 2024*; *Vukmirovi et al., 2025*). After rewatering, the Pn, Gs and Tr of each group were restored. Under LD stress, the photosynthetic parameters were even close to the control level, indicating that it had strong recovery ability after rewatering. However, under SD stress, the change trend of Ci was not obvious during the whole rewatering process, indicating SD-induced damage is difficult to fully recover by rewatering. As *Li et al. (2018)* studied in wheat, the growth of wheat was severely inhibited under severe drought stress. After rewatering, it took a long time for the plant to return to normal growth, and the recovery was slow and difficult to return to normal levels.

The change of MDA content in plants is closely related to the degree of oxidative damage of membrane lipids (*Jia et al., 2013b*). The content of MDA serves as a crucial indicator of cell membrane lipid peroxidation, reflecting the extent of membrane damage under stress. In this study, the content of MDA in all treatment groups initially increased and then decreased as drought stress progressed. And the MDA content of each treatment was higher than that of CK group. This indicates that severe drought stress will accelerate the process

of membrane lipid peroxidation, so that plants cannot effectively deal with the damage caused by drought, which is consistent with the results of previous studies (*Ke & Jin, 2007*). After rewatering, the MDA levels in all treatment groups decreased, indicating that some degree of oxidative damage to cell membranes persisted. The slower decline in MDA levels under MD and SD group compared to LD group suggests that severe stress conditions result in more prolonged and possibly irreversible membrane damage, thereby limiting the plant's recovery capacity. For example, studies have shown that drought stress can lead to lipid peroxidation of plant cell membranes and increase MDA content, which is a sign of cell membrane damage (*Mohammed, Kinet & Lutts, 2002*). However, this study found that the MDA content of the LD treatment group decreased after reaching the highest value on the 21st day, while other studies usually reported a continuous increase in MDA content during drought stress (*Mittler, 2002*). This difference may be due to plant species or drought intensity differences. The RC can indicate the degree of damage to the plant cell membrane. In this study, the RC of *M. faberi* leaves showed an upward trend and was significantly higher than that of the CK group, indicating that the cell membrane permeability of *M. faberi* leaves was destroyed with the extension of time, which was consistent with the results of *Anjum et al. (2011)* that drought caused cell membrane damage and increased RC. However, this study also has some unique findings. For example, we observed a significant decrease in the RC of the LD treatment group on the 28th day, while other studies have generally reported a continuous increase in RC under continuous drought stress (*Kocheva et al., 2004*). This phenomenon may be related to the self-repair mechanism of plants, such as alleviating further cell membrane damage by enhancing antioxidant enzyme activity (*Mittler, 2002*).

Under drought stress, the content of osmotic adjustment substances in plants generally increases (*Wang et al., 2000*; *Zhang et al., 2004*). In this study, the contents of SS, SP and Pro of drought treatment groups (LD, MD and SD) were generally higher than CK group at some time-points during drought stress period, consistent with the study of *Li, Peng & Su (2013)* on the physiological response of different leaf types of white clover to drought stress. The soluble sugar content showed a significant downward trend on the 28th day, and began to increase significantly after the plant was rewatered. This may be due to the prolonged drought stress leading to accelerated decomposition and synthesis of soluble sugar decreased (*Zhu, Wang & Xue, 2005*).

There is a set of active oxygen scavenging system in plants, and its main functions include maintaining the balance of active oxygen metabolism and protecting membrane structure (*Huang et al., 2019*). In this study, the activities of SOD, POD and APX in the leaves of *M. faberi* showed an increasing trend in the early stage of stress, which was consistent with the results of *Wang (2015)*. This variation trend suggests that ROS accumulation induces enzyme activity increase, balancing ROS dynamically. However, with the extension of time, the changes of SOD, POD and APX are different, indicating that long-term stress may induce changes in enzyme localization or regulation mechanism (*Du, Zhou & Huang, 2013*). Upon rewatering, SOD, POD, and APX activities in each treatment were decreased at least at one time-point, indicating that drought-induced antioxidant system damage gradually diminished post-rewatering. Numerous studies have shown that rewatering after

drought can rapidly restore plant growth, remove drought inhibition, and even lead to over-compensation, helping plants coping with stress (*Hao, Guo & Zhang, 2009*). This study shows that most indicators exhibit different degrees of compensation effect after rewatering.

The correlation analysis of the physiological and biochemical indexes of *M. faberi* under drought stress can reflect the physiological response of plants under drought stress, which plays an irreplaceable role in the application and promotion of plants. In this study, there was a significant positive correlation between photosynthetic indexes (Pn, Gs, and Tr) and chlorophyll content (Chlorophyll a, Chlorophyll a+b, and Chlorophyll a/b). The increase of chlorophyll content would lead to the increase of photosynthetic capacity of plants. This may be due to the fact that chlorophyll is an essential component of the photosynthetic apparatus, and its content directly affects the efficiency of photosynthesis (*Sharma et al., 2019*). However, there was a significant negative correlation between chlorophyll content and Pro, SP, and Ci content. This may be due to the fact that under drought stress conditions, plant photosynthetic apparatus is damaged, leading to a decrease in chlorophyll content and photosynthetic capacity. In order to maintain normal physiological functions, the plant accumulates Pro and SP as Osmo protectants to cope with the adverse effects of drought stress (*Krishankumar et al., 2025*; *Yan et al., 2024*).

This study highlights the critical role of antioxidant enzymes (SOD, POD and APX) and osmotic regulators (Pro and SS) in *M. faberi*'s drought tolerance. These biochemical mechanisms not only reduce ROS accumulation, thereby protecting cellular membranes from oxidative damage, but also help maintain osmotic balance and water retention, which are key to withstanding drought conditions and ensuring rapid recovery following LD and MD stress. Under rewatering conditions, the recovery of *M. faberi* was particularly significant after LD and MD stress. This recovery can be attributed to the plant's ability to dynamically adjust its physiological and biochemical mechanisms. Specifically, the enhanced activity of antioxidant systems during the early recovery phase likely facilitates the scavenging of ROS accumulated during drought stress, thereby mitigating oxidative damage and stabilizing cellular membranes. Furthermore, osmotic regulators such as Pro and SS likely play a pivotal role in the recovery process by helping maintain cellular turgor and promoting metabolic reactivation. The photosynthetic machinery, which experiences partial inhibition under drought stress, also demonstrates remarkable resilience,the rapid recovery of chlorophyll content and photosynthetic parameters such as Pn and Gs suggests that *M. faberi* possesses an efficient system for repairing or replacing damaged photosynthetic components. This aligns with findings in other species that photosystem repair and pigment resynthesis are critical for recovery after drought stress. By elucidating these recovery mechanisms, our study provides valuable insights into the biochemical and physiological basis of *M. faberi*'s resilience to drought stress. These findings emphasize the importance of integrated stress response systems in plants and suggest potential targets for enhancing drought tolerance through genetic or agronomic interventions.

## CONCLUSIONS

Taken together, this study demonstrated that drought stress significantly impacted the phenotype, photosynthesis system, and antioxidant system of *M. faberi*. The chlorophyll content and photosynthetic rate exhibited a downward trend, while the RC, MDA, antioxidant enzyme activity and osmotic adjustment substances showed an increasing trend. Additionally, a decrease in the density of leaf tissue arrangement suggested that the plants were continuously adapting their growth status and characteristics to manage stress. Rewatering treatments revealed that *M. faberi* recovered more effectively from LD and MD stress compared to SD conditions. These findings illustrate that *M. faberi* exhibits complex morphological, physiological and biochemical responses to drought, with a notable capacity for recovery following rewatering. However, under severe drought conditions, recovery is challenging, suggesting that it should not be planted in highly arid regions during cultivation management. Although this study provides insights for early drought tolerance, it only focuses on one-year-old seedlings, and these findings may not fully reflect the adaptability of mature plants. In order to understand the drought response of *M. faberi* more comprehensively, future research will focus on mature plants and evaluate their long-term response to continuous and intermittent drought cycles.

### Funding

This work was supported by the Natural Science Foundation of Hunan Province China (2024JJ5178), Changsha Natural Science Foundation of Hunan Province (kq2402112), the Forestry Science and Technology Innovation Foundation of Hunan Province for Distinguished Young Scholarship (XLKJ202205), the key project of the Hunan Provincial Education Department (22A0155), the Forest Bureau of Hunan Provence (XLKY2024), the Graduate Innovation Project of Hunan Province (2023XC108). The funders had no role in study design, data collection and analysis, decision to publish, or preparation of the manuscript.

### Grant Disclosures

The following grant information was disclosed by the authors:
The Natural Science Foundation of Hunan Province China: 2024JJ5178.
Changsha Natural Science Foundation of Hunan Province:  kq2402112.
The Forestry Science and Technology Innovation Foundation of Hunan Province for Distinguished Young Scholarship: XLKJ202205.
The key project of the Hunan Provincial Education Department:  22A0155.
The Forest Bureau of Hunan Provence: XLKY2024.
The Graduate Innovation Project of Hunan Province:  2023XC108.

### Competing Interests

The authors declare there are no competing interests.

## Author Contributions

- Wei Tang conceived and designed the experiments, performed the experiments, prepared figures and/or tables, and approved the final draft.
- Qiong Mo conceived and designed the experiments, performed the experiments, prepared figures and/or tables, and approved the final draft.
- Yangyang Fu analyzed the data, prepared figures and/or tables, and approved the final draft.
- Damao Zhang performed the experiments, analyzed the data, prepared figures and/or tables, and approved the final draft.
- Yang Liu performed the experiments, analyzed the data, prepared figures and/or tables, and approved the final draft.
- Mingtong Ren performed the experiments, analyzed the data, prepared figures and/or tables, and approved the final draft.
- Tingting Li conceived and designed the experiments, prepared figures and/or tables, and approved the final draft.
- En Wu analyzed the data, prepared figures and/or tables, and approved the final draft.
- Dingding Su conceived and designed the experiments, authored or reviewed drafts of the article, and approved the final draft.
- Xiaoying Yu conceived and designed the experiments, authored or reviewed drafts of the article, and approved the final draft.
- Lihong Yan conceived and designed the experiments, authored or reviewed drafts of the article, and approved the final draft.
- Yanlin Li conceived and designed the experiments, authored or reviewed drafts of the article, and approved the final draft.

## Data Availability

Raw data is available in the Supplemental Files.

## Supplemental Information

Supplemental information for this article can be found online at http://dx.doi.org/10.7717/peerj.19855#supplemental-information.

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
