# Peer review of "Physiological response mechanism of Machilus faberi Hemsl under drought stress and rewatering"

_PeerJ, doi:10.7717/peerj.19855_

## Round 0.1 · original submission · Major Revisions

The manuscript must be significantly improved in terms of language, clear hypothesis, unambiguous methodology, lengthy discussion, etc.

**Language Note:** The Academic Editor has identified that the English language must be improved. PeerJ can provide language editing services - please contact us at [email protected] for pricing (be sure to provide your manuscript number and title). Alternatively, you should make your own arrangements to improve the language quality and provide details in your response letter. – PeerJ Staff

Reviewer 1 ·

Basic reporting

English proficiency and grammar need to be improved for better clarity and readability of the manuscript.
Sufficient literature and background information were studied and provided.

Experimental design

NO Comments.

Validity of the findings

The work provides significant insights into Machilus faberi's drought stress response, however, it would be beneficial to highlight how these findings complement to previous research on drought resistance in tree species. What particular physiological traits distinguish M. faberi's response?
"The conclusions are well stated; however, a future direction for the study may be included in the conclusion section."
Given the ecological and economic importance of M. faberi, how may the findings guide conservation efforts or afforestation projects during climate change scenarios?
Are the physiological reactions species-specific, or might these findings apply to other Lauraceae species? this might be discussed.
T14 showed a significant reduction in shoot number in all the treatments including the control? Discuss the mechanism for this findings.
Are several rewatering cycles tested to simulate persistent droughts that occur in real-world conditions?

Additional comments

General Comments
Line no: 56 Scar-city may change to scarcity.
Line no: 58 have placed may change to have put
Line no: 60 change plants’ to plant
Line no: 61 Change roles to importance
Line no: 70 replace the sentence of “Water deficit leads to water imbalance in plants. Plants reduce water loss by closing stomata, but this also limits the absorption of carbon dioxide and reduces the efficiency of photosynthesis(Zhang et al. 2011)” as “Water deficit impairs the water balance in plants. To reduce water loss, plants close their stomata; however, this also limits carbon dioxide absorption and reduces photosynthetic efficiency."
Line no : 79 change so as to - “To”
Line no: 89 Yang et al., the year is missing
Line no: 95 im-portance to importance
Line no: 110 gar-den to garden
Line no:118-119 cite reference for pot water control method.
Line no: 121-122 Use the Equation function in MS Word for better clarity when presenting formulas
Soil moisture content= (Wet weight-Dry weight)/(Dry Weight) ×100 %
Line no: 130 After of the drought treatment may change to After the drought treatment
Line no: 165 accurate to an accurate
Use consistent word choices throughout the manuscript; write 'minutes' or 'mins' uniformly to maintain integrity in unit usage.
Line no: 275 decreased instead of decreasing
Line no: 282 initial instead of initially
Line no:294, 311, 322,329, etc., The abbreviation 'Gs' for stomatal conductance will be introduced when it first appears in the manuscript. Please apply this rule to all abbreviations throughout the manuscript."
Line no:365 Replace the water- deficient to “a water- deficient”
Line no: 384 Replace on the 49th day to “from” the 49th day
Line no:389 The change trend of palisade tissue to The “changing” trend

Reviewer 2 ·

Basic reporting

The paper focuses on a very well studied stress condition on plants: drought. The novelty should be the less known specific responses in Machilus faberi, studied by applying different levels of soil moisture followed by re-watering. However, the introduction lacks a clear hypothesis and is oriented towards a generic description of responses without any structured idea. This lack becomes very important in the discussion, where there is too wide description of the results without a clear link between them. In fact, the results obtained by comparing different levels of stress are weakly discussed and almost completely lost in the conclusion. In the results section, the description is often inconsistent with the graphs, leading to incorrect interpretation. In fact, any speculative or debating sentences need to be removed from the results and eventually placed in the discussion section. English sometimes needs to be improved. The reference style must be checked (there is always a space missing) and the figures in the figure panel must be compared in the same order as those mentioned in the test.

Experimental design

The experimental design is clear.

Validity of the findings

This is the weak point of the paper, the authors made several analyses, but in the paper the connection, the significance and the impact of the results are completely missing, although they wrote a long discussion. The reason for this may be the lack of a clear hypothesis, which turns the paper into a broad description of data from different areas of analysis without any structured comparison between them.
The description of the results is not always correct and corresponds to the graphs, which causes a lot of confusion in the discussion section. Then I wonder why you monitored one CK group and never used it for comparison?

Additional comments

Please find attached the pdf of the paper with some comments that you can use to improve your work.

Annotated reviews are not available for download in order to protect the identity of reviewers who chose to remain anonymous.

---

## Round 0.2 · Major Revisions

Authors have revised the manuscript significantly. However, they have to focus on writing part. Revise the whole manuscript as per comments of the reviewers.

**Language Note:** The review process has identified that the English language must be improved. PeerJ can provide language editing services - please contact us at [email protected] for pricing (be sure to provide your manuscript number and title). Alternatively, you should make your own arrangements to improve the language quality and provide details in your response letter. – PeerJ Staff

Reviewer 1 ·

Basic reporting

Authors have revised the manuscripts carefully based on the previous comments.

Experimental design

No comments.

Validity of the findings

Significantly improved based on the previous comments.

Additional comments

Still, in some sections, English and grammar mistakes need to be corrected. Carefully check throughout the manuscript.
Briefly relate how climate change increases the urgency for such targeted planting and conservation strategies?
Line No.155-157 change the line into “Plant height was measured using a tape measure from the soil surface of the potted plant to the top of the main stem. 'Shoots' refer to the new growth emerging from the main stem after pruning."
Line No: 163 determined using an ultraviolet spectrophotometer (UNICO, Shanghai, China) instead of measured.
Line no: 176-177 Change the line to "The measurements were taken from a 6 cm² area of a leaf from a one-year-old M. faberi seedling."
Line no: 204 Write as “An inverted microscope”
Line no: 258 change to reaching a peak
Line no: 259 Chlorophyll a ranges
Line no: 288 LD and MD groups
Line no: 335 change the line to reach a peak
Line no: 336 change to ranges
Line no:362 LD and MD groups gradually increases
Line no: 405 Remove basically
Line no: 537 change to gradually recovered
Line no: 541 change the line as “Meanwhile, with the extension of drought time, the plant height and new branch length of M.faberi decreased gradually.
Line no: 544 change the line as “It shows that drought leads to a decrease in soil moisture and difficulty in water absorption by plant roots, which leads to the slow growth of aboveground parts (Mahmood et al. 2025)”.
Line no: 729 Osmoprotectants

Reviewer 2 ·

Basic reporting

The authors have worked on and improved the manuscript according to the suggestions, but there are still many mistakes in the text that need to be corrected before the work can be published.

Please see the "Additional comments" section to see my comments.

I kindly ask the authors to pay attention to the description of the results (both in the results and in the discussion sections), as there are still errors in the order of the figures and in the presentation of the results.

Finally, please note the text form. There are many missing spaces or dots in the wrong place.

Experimental design

Nothing to add.

Validity of the findings

Nothing to add.

Additional comments

Abstract

There should be no explanation of the abbreviation in the abstract. So, you should use the full name of the parameter directly, e.g. stomatal conductance, or the abbreviation, e.g. gs, but not both (stomatal conductance (gs)).

Introduction

The introduction has been improved but still lacks a clear hypothesis. Even if you have clarified the scope of the work you have done, you have not hypothesised about the finding you are looking for. Please add the hypothesis you had when planning the experiment.

Materials and methods

Line 170. We did not understand each other in my previous comment. Here you say that you multiply the results R1/R2 per 100, but in the results section it is the ratio and not the percentage. Please correct both the material and method and the title of the Y-axis in the graph. Alternatively, you could multiply all RC values by 100 and leave the text unchanged.

Line 197. Change "P" in p-value, with p in italics.

Results

Line 215. When you say "significant" differences, you should be referring to a statistical test. As there is no statistic for phenotype and growth measures, please change significant to evident or something similar.

Lines 258-260. "Significantly lower" is statistically incorrect. What the statistical test highlights is whether or not two groups are different, without inferring anything about the 'history' of the samples' rise or fall. Also, I wonder how the MD and SD can be statistically similar (both with the letter "c") when the means are so different and the standard deviations so small. Please check if the statistics of these samples were done correctly. After all, SD was still decreasing at WT7 and LD and CK were not different. Please correct the text.

Line 261. "showed a brief increase from T7 to T14", I do not see the same. Please remove this text.

Lines 262-265. The description of the results here is very chaotic and lacks meaning. I suggest looking at the difference between the stressed group and CK at each time point. By doing this, until T14 it can be said that there is no difference due to drought stress.The differences compared to CK appeared at T21 for MD and SD and at T35 for LD. At WT21 all groups recovered well.

Lines 271-272. “the MD and SD groups…” I am sorry, but that is not true or well expressed.

Lines 273-274. “And the RC…” I am sorry, but that is not true or well expressed.

Line 277. For calculating the fold change, did you compare the treated group with the CK at the same time or with the CK at t0? I think you should use the value of CK at the same time point to be sure to capture the difference due to drought stress and avoid comparing it with t0: as CK is variable, other factors (growth, temperature,...) could probably have influenced the measured parameter.

Line 279. nmol/g FW as you wrote in the text or nmol/mg FW as you showed in the graph? Please correct. Also, when you are studying the effects of drought stress, it is dangerous to use fresh weight because it is strongly influenced by stress. Do you have some data on relative water content to do the calculation and express the results in dry weight?

Line 284. The first graph you showed (Fig. 4A) is SOD not POD, please correct the order.

Line 286. Even though they peaked on day 56, they already started to differ from CK on T49. Please highlight this.

Line 286. See line 258 comment.

Lines 289-290. “After rewatering, POD…” I am sorry, but that is not true or well expressed.

Lines 291-294. The comparison between drought levels is missing: please point out that SD already differed from CK at T35, whereas the other stressed group only started at T49.

Lines 297-298. See the comment to the line 277.

Line 299-300. “After rewatering, APX…” it increased also in CK so maybe it is not due to the stress.

Line 308. “and there was no significant difference with the Ck group” I am sorry, but that is not true or well expressed.

Lines 310-311. “Which was 2.3 times higher than that of the CK group”. See the comment to line 277.

Line 318. “In contrast” In fact, I can see no difference between the increment of Pro in MD and SD, except for the time point.

Lines 320-321. “with levels on the 14th…” I am sorry, but that is not true or well expressed.

Discussion

Line 379. Replace “With” with “with”.

Line 382. Please remove “it shows that”.

Line 386. See the comment to the line 258.

Line 402. Populus alba x Populus berolinensis in italic.

Line 414. Please, replace “is still ongoing” with “was still ongoing”.

Line 415 and 417. "CO2" with subscript 2.

Line 424. “Li Yanbin et al.” Please, insert the ref.

Line 458. See the comment to the line 258.

Line 468. Please check the comma between SOD and POD.

Lines 473-475. “Upon rewatering…” I am sorry, but that is not true or well expressed.

---

## Round 0.3 · accepted · Accept

The authors have addressed all the comments, and the reviewer is satisfied with the revised version of the manuscript. Therefore, the manuscript can be accepted for publication.

Reviewer 2 ·

Basic reporting

According to the previous comments, the authors reviewed the text. The manuscript is ready for publication.

Experimental design

I have no other comments to add.

Validity of the findings

I have no other comments to add.

Additional comments

I have no other comments to add.